# The Full Informational Spectral Analysis for Auditory Steady-State Responses in Human Brain Using the Combination of Canonical Correlation Analysis and Holo-Hilbert Spectral Analysis

**DOI:** 10.3390/jcm11133868

**Published:** 2022-07-04

**Authors:** Po-Lei Lee, Te-Min Lee, Wei-Keung Lee, Narisa Nan Chu, Yuri E. Shelepin, Hao-Teng Hsu, Hsiao-Huang Chang

**Affiliations:** 1Department of Electrical Engineering, National Central University, Taoyuan 320, Taiwan; temin.lee@gmail.com (T.-M.L.); fifaworld91@g.ncu.edu.tw (H.-T.H.); 2Department of Rehabilitation, Taoyuan General Hospital, Taoyuan 330, Taiwan; waikeung@mail.tygh.gov.tw; 3CWLab International, Thousand Oaks, CA 91320, USA; narisa.chu@ieee.org; 4The Pavlov Institute of Physiology, Russian Academy of Sciences, 199034 St. Petersburg, Russia; yshelepin@yandex.ru; 5Division of Cardiovascular Surgery, Taipei Veterans General Hospital, Taipei 112, Taiwan; 6Department of Surgery, School of Medicine, Taipei Medical University, Taipei 106, Taiwan

**Keywords:** auditory steady-state response (ASSR), electroencephalography (EEG), canonical correlation analysis (CCA), Holo-Hilbert spectral analysis (HHSA)

## Abstract

Auditory steady-state response (ASSR) is a translational biomarker for several neurological and psychiatric disorders, such as hearing loss, schizophrenia, bipolar disorder, autism, etc. The ASSR is sinusoidal electroencephalography (EEG)/magnetoencephalography (MEG) responses induced by periodically presented auditory stimuli. Traditional frequency analysis assumes ASSR is a stationary response, which can be analyzed using linear analysis approaches, such as Fourier analysis or Wavelet. However, recent studies have reported that the human steady-state responses are dynamic and can be modulated by the subject’s attention, wakefulness state, mental load, and mental fatigue. The amplitude modulations on the measured oscillatory responses can result in the spectral broadening or frequency splitting on the Fourier spectrum, owing to the trigonometric product-to-sum formula. Accordingly, in this study, we analyzed the human ASSR by the combination of canonical correlation analysis (CCA) and Holo-Hilbert spectral analysis (HHSA). The CCA was used to extract ASSR-related signal features, and the HHSA was used to decompose the extracted ASSR responses into amplitude modulation (AM) components and frequency modulation (FM) components, in which the FM frequency represents the fast-changing intra-mode frequency and the AM frequency represents the slow-changing inter-mode frequency. In this paper, we aimed to study the AM and FM spectra of ASSR responses in a 37 Hz steady-state auditory stimulation. Twenty-five healthy subjects were recruited for this study, and each subject was requested to participate in two auditory stimulation sessions, including one right-ear and one left-ear monaural steady-state auditory stimulation. With the HHSA, both the 37 Hz (fundamental frequency) and the 74 Hz (first harmonic frequency) auditory responses were successfully extracted. Examining the AM spectra, the 37 Hz and the 74 Hz auditory responses were modulated by distinct AM spectra, each with at least three composite frequencies. In contrast to the results of traditional Fourier spectra, frequency splitting was seen at 37 Hz, and a spectral peak was obscured at 74 Hz in Fourier spectra. The proposed method effectively corrects the frequency splitting problem resulting from time-varying amplitude changes. Our results have validated the HHSA as a useful tool for steady-state response (SSR) studies so that the misleading or wrong interpretation caused by amplitude modulation in the traditional Fourier spectrum can be avoided.

## 1. Introduction

Steady-state response (SSR) elicited by periodic sensory stimuli is a synchronized and phase-locked type of neural activity in the human brain. The SSR has been extensively used to investigate the electrophysiological responses underlying different neural networks [1]. Different stimulus parameters, such as stimulus frequency [2], duty cycle [3], intensity contrast [4], phase synchrony [5], attention modulation [6], etc., have been studied to characterize their functional capacities and neural mechanisms. Classical physiological models assume SSR as a stereotypical time-invariant neural response, in which background fluctuations are linearly added to the elicited SSR. A time-averaging technique is usually chosen to suppress the signal fluctuations that are uncorrelated to the given periodic stimulus. However, recent studies indicate SSR contains nonlinear neural responses, which could be modulated by the subject’s attention [7], wakefulness state [8,9], mental load [10,11,12,13], and mental fatigue [14]. Traditional frequency analysis methods, such as Fourier transform or wavelet transform, assume the measured time series are the linear combinations of vectors from a predetermined basis, in which the modulation among different signal scales is not considered. Nevertheless, it has been demonstrated that neural adaption can alter the strategy of neural coding and cause modulation in the phase, latency, or amplitude in evoked potentials [15]. Several studies have shown neural coding plays an important role in the modulation of oscillatory activities within cortical neuron populations [16]. The neural coding describes that the information transmitted from sensory perception to the central nervous system can be encoded by varying the rate and timing of action potentials (spikes), depending on the properties of the perceived stimulus and the subject’s physical states [17]. For example, Zhong et al. (2013) investigated the effects of noise-induced hearing loss on the coding of an envelope structure in the central auditory system, and they observed increased auditory responses at the applied modulation frequency in people with cochlear hearing loss [18]. Nasir et al. (2013) tested the plasticity in sensory systems using somatosensory evoked potentials (SEPs) and observed that sensorimotor adaption induced alternation in the neural coding of somatosensory stimuli [19]. Several neural coding hypotheses have been proposed to account for the processing of neural activity patterns in our brain and nervous systems, such as rate coding, temporal coding, population coding, correlation coding, independent spike coding, position coding, sparse coding, etc. [20]. Among the neural coding hypotheses, rate coding and temporal coding are the two neural coding strategies that were studied the most in previous literature. The rate coding model, sometimes called the frequency coding model, hypothesizes that the information of a signal is contained in the firing rate of neural spikes, while the temporal coding model assumes the information of interneuron transmission is contained in the precise timing of spikes and inter-spike intervals. Aghababaiyan et al. (2019) studied the capacity and error probability analysis of neuro-spike communication using a numerical simulation approach and concluded temporal coding has a higher efficiency than spike rate coding in terms of achievable data rate [21]. The neural coding could also be influenced by a subject’s physiological states, such as attention [22] and emotion states [23], in order to achieve better efficiency of information transmission [24]. Since the exertion of a modulation frequency on a sinusoidal signal will result in a frequency shifting away from its oscillatory frequency according to the trigonometric product-to-sum formula, neural modulation, induced by changes in internal statuses or external environments [25], could result in deviation in the detected oscillatory frequency of human SSR. The development of a frequency analysis method, which takes both the oscillatory and modulatory frequencies into account, is necessary to study the subtle dynamics in the measured SSR.

Auditory steady-state responses (ASSR) elicited by rapid auditory stimuli are a kind of SSR commonly used in clinics. ASSR is a sinusoidal-like response consisting of evoked responses from the central auditory pathway and auditory cortex, which allows the measurement of hearing response at a specified frequency. The modulation technique for ASSR stimulus can be either amplitude modulation [26,27,28] or frequency modulation [29]. It has been demonstrated that ASSR is a useful translational biomarker for several neurological and psychiatric disorders [30,31,32,33]. The ASSR is also an objective tool for estimating hearing sensitivity in individuals, especially for hearing loss [34,35], hearing assessment [36,37], and evaluation of aural rehabilitation [38]. Khuwaja et al. (2015) measured the ASSR responses during different stages. They obtained a 100% classification rate for sleep stage classification using a neural network classifier [39]. Swanepoel (2011) compared ASSR and brain stem response (ABR) in infants and concluded ASSR is a more reliable and useful tool for diagnosing hearing loss in infants [40]. Sugiyama et al. (2021) reviewed the ASSR processing in psychiatric disorders, including schizophrenia, bipolar disorder, and autism. The ASSR amplitude is suppressed within the gamma band (~40 Hz), which indicates the imbalance between GABAergic and N-methyl-D-aspartate (NMDA) receptor-medicated neurotransmission [41]. Yokota and Naruse (2015) found that the phase coherence of ASSR can reflect the amount of cognitive workload in an N-back task [42]. Kallenberg et al. (2007) implemented an ASSR-based brain–computer interface (BCI) by giving two tones with distinct modulation frequencies. Subjects were requested to pay attention to one out of the two tones to achieve the auditory BCI control [43].

Most ASSR studies utilized tone bursts modulated at 40 Hz, in which the test frequencies are often chosen at 500, 1000, 2000, and 4000 Hz with a 100% modulation depth [38]. Physicians adjust the intensities of ASSR auditory stimuli from high to low to objectively determine the subject’s hearing threshold. In most of the aforementioned studies, the ASSR were detected using ensemble average or Fourier-based spectral analysis. Both the techniques assume the auditory-induced responses are stationary across the whole measurement data. Temporal modulation in the recorded auditory response is usually neglected, which could result in the information loss for studying ASSR neural mechanisms. 

In order to preserve the dynamic information of ASSR, several signal processing approaches have been proposed to suppress ASSR-unrelated noise in the EEG recordings. Biesmans et al. (2015) filtered the EEG signals within a narrow band and designed a spatial filter to enhance the neural activities by optimizing the SNR of recorded ASSR [44]. Janani et al. (2018) utilized independent component analysis (ICA) to suppress ASSR-unrelated noise by removing artifact-contaminated components. They proposed a correlation-based algorithm for the component selection process [45]. Wang et al. (2015) applied empirical mode decomposition (EMD) to extract ASSR signals in normal subjects and tinnitus patients [46]. They constructed a spatial map for each intrinsic mode function (IMF), and the ASSR components were identified by comparing the spatial map distribution with a pre-defined spatial template. Bin et al. (2009) [47] applied canonical correlation analysis (CCA) to enhance the steady-state visual evoked potential (SSVEP)-associated frequency information, and they used it as the control signal for brain–computer interface (BCI) control. The aforementioned methods utilized spatial filter or blind source separation (BSS) to achieve noise removal in EEG signals. However, these methods did not analyze the effect of time changes in amplitudes, in which the temporal modulation can lead to frequency splitting or spectrum broadening (trigonometric product-to-sum formula). The frequency shift caused by amplitude modulation will result in ambiguity in clinical ASSR diagnoses. 

One spectral analysis tool, the Holo-spectral analysis (HHSA) proposed by Huang et al. (2016) [48], was proposed to analyze both the amplitude modulation (AM) and the carrier frequency (frequency modulation, FM) information in a measured signal. The HHSA was constructed based on the structure of two-layer empirical mode decompositions (EMD), in which the EMD is a signal decomposition method used to extract meaningful oscillation features into intrinsic mode functions (IMF) by means of recursively applying a sifting process [49]. In HHSA, the first-layer EMD decomposes the input signal into frequency modulation (FM) components. In the second layer of HHSA, the signal envelopes of the IMFs obtained from the first-layer EMD are further decomposed into amplitude modulation (AM) components. The HHSA enables a two-dimensional AM-FM representation of the input signal, in which the FM frequency represents the fast-changing intra-mode frequency, and the AM frequency represents the slow-changing inter-mode frequency. In this study, we propose a method by combining the CCA and HHSA for the spectral analysis of ASSR. The CCA was applied as a signal preprocessing approach by selecting ASSR-related canonical vectors. The ASSR-related canonical vectors were chosen for reconstructing noise-suppressed channel signals. The source activities at the left auditory cortex (LAC) and right auditory cortexes (RAC) were estimated, and the HHSA was applied to give the AM-FM representation of the measured ASSR. In our analysis, we adopted CCA as preprocessing for noise removal and applied the HHSA for AM-FM analysis. The study results might shed light on exploring the subtle information processing underlying the auditory neuronal circuitry.

## 2. Materials and Methods

### 2.1. Auditory Stimulation

Our steady-state auditory stimuli utilized a 1 kHz sinusoidal wave modulated at 37 Hz with 100% modulation depth [50]. Monaural auditory stimuli were presented to the left and right ears of each participant in separate sessions. Each session was three minutes in duration, and triggers were generated at the beginning of each second for subsequent signal processing. The intensities of auditory stimuli were set to 70 dB sound pressure level (SPL), generated by an analog-to-digital conversion card (D/A) conversion card (NI USB-6259, National Instrument, Austin, TX, USA), which was precisely controlled by the LabView software (National Instruments, Austin, TX, USA). 

### 2.2. Subjects and Tasks

Twenty-five normal subjects (14 males and 11 females, all right-handed subjects; mean ages = 31 ± 4.2 years) were recruited to participate in this experiment. Subjects were requested to sit in a comfortable armchair in a dimly illuminated electromagnetic shielded room. All participants were requested to participate in two auditory stimulation sessions, including one right-ear and one left-ear monaural steady-state auditory stimulations (one for the right ear and one for the left ear) for three minutes. A three-minute empty-room measurement was also recorded for each participant to monitor the background noise. All participants gave informed consent, and the study was approved by the Ethics Committee of the Institutional Review Board (IRB), Tao-Yuan General Hospital, Taiwan. The measurements were noninvasive, and the subjects were free to withdraw at any time. 

### 2.3. Electroencephalography Recordings

The electroencephalography (EEG) signals were recorded using a 32-channel whole-head EEG system (band-pass, 0.05 Hz–250 Hz; digitized at 1 kHz; QuickAmp, Brain Products Co., Munich, Germany). A 46 Hz–65 Hz bandstop filter (3rd-order Butterworth IIR filter) was also applied to filter out electricity noises, in which the 46 and 65 Hz cut-off frequencies were chosen by setting 9 Hz frequency margins for the ASSR-related 37 Hz (fundamental stimulation frequency) and 74 Hz (the first harmonic frequency) frequencies, respectively. Electrodes were placed in accordance with the international 10–20 channel placement. The exact position of the head with respect to the EEG electrodes was determined by measuring magnetic signals from four head position indicator (HPI) coils placed on the scalp. Coil positions were identified with a three-dimensional digitizer (Fastrack system, Polhemus, Colchester, CT, USA) with respect to three predetermined landmarks (naison and bilateral preauricular points) on the scalp, and these data were used to superimpose EEG source signals on individual MRI images, obtained with a 3.0 T Bruker MedSpec S300 system (Bruker, Kalsrube, Germany). The anatomical image was acquired using a high-resolution T1-weighted, 3D gradient-echo pulse sequence (MDEFT: Modified Driven Equilibrium Fourier Transform; TR/TE/TI = 88.1 ms/4.12 ms/650 ms, 128 × 128 × 128 matrix, FOV = 250 mm). The whole-head EEG signals recorded in monaural steady-state auditory stimulations were stored on a hard disk for subsequent off-line CCA and HHSA processing.

### 2.4. Canonical Correlation Analysis and Selection Pertinent 

Since the EEG channels in temporal areas are close to the human auditory cortex, the channels from left or right temporal areas (the F7, FC5, T7, and CP5 in the left auditory area; the F8, FC6, T8, and CP6 in the right auditory area) were chosen as channels-of-interest (COI) for ASSR extractions. We considered the ASSR responses generated from the left auditory area might not be synchronized with the ASSR responses obtained from the right auditory area. The ASSR responses from the left auditory area and right auditory area were extracted by setting the COI at left and right auditory areas separately. Given an *M*-channel and *N* time points for EEG measurement X*_M_*_×*N*_ (*M* = 4, including the F7, FC5, T7, and CP5 in the left auditory area, or the F8, FC6, T8, and CP6 in the right auditory area) and a set of reference signals Y*_K_*_×*N*_ (*K* = 4 in this paper, containing the sinusoidal signals with fundamental ASSR frequency and its first harmonic frequency) with *N* time points (*N* = 180,000), the canonical correlation analysis (CCA) tries to find vectors *a* ($a\in {ℜ}^{M})$ and *b* ($b\in {ℜ}^{K})$ to maximize the correlation $\rho =corr(a^{T}X,b^{T}Y)$, in which the *u* = *a^T^*X and *v* = *b^T^*Y are defined as one pair of canonical vectors. The CCA seeks to find *z* (*z* = min(*M*,*K*)) pairs of canonical vectors, and the *i*th pair of canonical vectors are uncorrelated with the *j*th pair of canonical vectors (i.e., $u_{i}⊥u_{j}$ and $v_{i}⊥v_{j}$, for *i* ≠ *j*). In this study, we took the fundamental frequency and the first harmonic frequency of steady-state auditory stimulus into consideration so that the reference signals Y can be represented as follows: (1)$$Y=\left[\begin{array}{c} \sin(2\pi \times f_{0}\times t)\\ \cos(2\pi \times f_{0}\times t)\\ \sin(2\pi \times 2\times f_{0}\times t)\\ \cos(2\pi \times 2\times f_{0}\times t) \end{array}\right],$$
where *f*_0_ (*f*_0_ = 37 Hz) is the ASSR stimulus frequency and *t* is the time index. 

The correlation value between the *i*th pair of canonical vectors is: (2)$$\rho_{i}=\frac{a_{i}^{T}\sum_{\mathrm{XY}}b_{i}}{\sqrt{a_{i}^{T}\sum_{\mathrm{XX}}a_{i}}\cdot \sqrt{b_{i}^{T}\sum_{\mathrm{YY}}b_{i}}},$$
where $\rho_{i}$ is the correlation value between *u_i_* and *v_i_*, *u_i_* = *a_i_^T^*X and *v_i_* = *b_i_^T^*Y are the *i*th pair of canonical vectors, $\sum_{\mathrm{XX}}$ and $\sum_{\mathrm{YY}}$ are the auto-correlation matrixes of X and Y, $\sum_{\mathrm{XY}}$ is the cross-covariance matrix between X and Y. 

The *u_i_* (*i* = 1, …, 4) constructs a signal space obtained from the original EEG recordings around the left and right auditory area so that the canonical vectors are the vector directions which multichannel EEG samples correlate with most. The *v_i_* (*i* = 1, …, 4) are the signal spaces constructed from the reference signals, which are not used in this study because we want to reconstruct the ASSR responses based on the signal features extracted from original recordings, rather than the signal space constructed from the reference signals. Further arranging of the vectors for creating each pair of canonical vectors into matrix A, the four canonical vectors can be obtained by the multiplication of X and A, represented as: (3)$$U_{4\times N}={\left(A_{4\times 4}\right)}^{T}\cdot {\left(X_{4\times N}^{}\right)}^{}={\left[\begin{array}{c} U(1)\\ U(2)\\ U(3)\\ U(4) \end{array}\right]}^{},$$
where *U* contains the four 1×N canonical vectors, $A=\left[\begin{array}{cccc} a_{1} & a_{2} & a_{3} & a_{4} \end{array}\right]$ contains the matrix to create the four 1×N canonical vectors, and *U*(1), *U*(2), *U*(3), *U*(4) are the canonical vectors generated from the EEG recordings in the first, second, third, and fourth pairs of canonical vectors, respectively. 

Figure 1 shows the spatial maps and Fourier spectra of the four canonical vectors. The spatial map was created to represent the topographic distribution (correlation) of each canonical vector over the whole-head EEG channels. For the 32-channel whole-head EEG recording E_32×*N*_, the correlation of each canonical vector on all EEG channels can be calculated by: (4)$$M{\left(i\right)}_{32\times 1}^{}=ABS\left\{{\left(E_{32\times N}\right)}^{}\cdot {\left(U{\left(i\right)}_{1\times N}/\left\|U{\left(i\right)}_{1\times N}\right\|\right)}^{T}\right\},$$
where E_32×*N*_ contains the 32-channel EEG recording samples with *N* time points, the *U*(*i*) is the *i*th canonical vector obtained by setting COIs surrounding the left or right auditory area, ‖ ‖ is the function of the L2-norm, *ABS*{·} is the function of absolute value, and *M*(*i*) contains the absolute values of the correlation coefficients between the *i*th canonical vector and E_32×*N*_. In CCA, the relation between each canonical vector and the reference signals is examined by performing a statistical significance analysis. The canonical vectors with *p* < 0.01 were identified as ASSR-related canonical vectors and were subjected to the reconstruction of noise-suppressed ASSR responses. Figure 1 demonstrates the CCA processing for extracting ASSR responses in subject 5. The left panel (Figure 1A) shows the channel plot of the recorded EEG signals. The middle-upper panel (Figure 1B) and the middle-lower panel (Figure 1C) show the spatial maps and the Fourier spectra of canonical vectors obtained by setting COIs located around left and right temporal areas. In Figure 1B, the canonical vectors *U*(1) and *U*(2), whose *p* values are smaller than 0.01 (*p* < 0.01) and had clear 37 Hz spectral peaks with their spatial maps concentrated on the right temporal area, were chosen as ASSR-related canonical vectors of the right hemisphere. Similarly, in Figure 1C, the *U*(1), *U*(2) and *U*(3) (*p* < 0.01) obtained from left COIs, which presented clear 37 Hz spectral peaks with spatial maps concentrated on the left temporal area, were identified as ASSR-related canonical vectors of the left hemisphere. The EEG signals were then projected to the canonical vectors obtained from left and right ROIs, and the signals were reconstructed to suppress ASSR-unrelated noises. The data reconstruction procedure can be represented by the following equation: (5)$$E_{32\times N}^{recon}=\left\{\sum_{\begin{array}{l} S\mathrm{elected}\;i\;\mathrm{from}\;\\ \mathrm{canonical}\;\mathrm{vectors}\\ \mathrm{of}\;\mathrm{left}\;\mathrm{ROI} \end{array}}\left[{\left(E_{32\times N}\right)}^{}\cdot {\left(U_{L}{\left(i\right)}_{1\times N}/\left\|U_{L}{\left(i\right)}_{1\times N}\right\|\right)}^{T}\right]\cdot U_{L}(i)\right\}+\left\{\sum_{\begin{array}{l} S\mathrm{elected}\;i\;\mathrm{from}\;\\ \mathrm{canonical}\;\mathrm{vectors}\\ \mathrm{of}\;\mathrm{right}\;\mathrm{ROI} \end{array}}\left[{\left(E_{32\times N}\right)}^{}\cdot {\left(U_{R}{\left(i\right)}_{1\times N}/\left\|U_{R}{\left(i\right)}_{1\times N}\right\|\right)}^{T}\right]\cdot U_{R}(i)\right\},$$
where *E_recon_* is the reconstructed ASSR-related EEG recordings, *E* is the original EEG recording, and *U_L_* and *U_R_* are the canonical vectors obtained by setting COIs close to the left and right temporal areas. The ASSR-related EEG signals were reconstructed by examining ASSR-related canonical vectors (*p* < 0.01) and the original EEG recordings were projected to these ASSR-related canonical vectors to suppress task-unrelated components. It can be noticed that the spatial maps generated from ASSR-related canonical vectors on the left and right hemispheres had the characteristic of unilateral distributions, which indicated the low inter-hemispheric correlation of ASSR responses. Accordingly, in this study, we first calculated the ASSR responses in the left and right hemispheres separately and then summated them together to obtain the whole-head ASSR responses (Figure 1D) for the following ASSR source estimations.

### 2.5. Analysis of ASSR Source Activities Using Minimum Norm Estimation

To estimate the source activities in the left and right auditory cortexes (LAC and RAC), minimum norm estimation (MNE) with a realistic model generated from an individual’s magnetic resonance image (MRI) (BrainStorm software, University of South California, Los Angeles, CA, USA; http://neuroimage.usc.edu/brainstorm, accessed on 11 December 2021) was adopted [51]. Only the nodes on the cortical surface with source amplitudes survived statistical significance (*p* < 0.01) among the total surface nodes that were rendered on an individual’s anatomical MRI (see Figure 2). Figure 2A,B shows the source estimation results obtained from the reconstructed EEG signals shown in Figure 1D. The auditory-induced source activities were estimated by means of minimum norm estimation (MNE) (BrainStorm software, University of South California, Los Angeles, USA; http://neuroimage.usc.edu/brainstorm, accessed on 11 December 2021) [51], with a realistic head model generated from an individual’s magnetic resonance image (MRI) using brainVISA software (SHFJ, Orsay, France; http://brainvisa.info/, accessed on 11 December 2021) [52]. The estimated neural sources were overlaid on anatomical MRI, and only those cortical surface nodes with source amplitudes that survived statistical significance (*p* < 0.05) among total surface nodes were rendered on MRI. The source activities in RAC and LAC were plotted in Figure 2C,D, respectively. Figure 2E,F shows the power density spectra (Welch’s power spectrum [53]; signal window 1000 ms; overlapping 500 ms) obtained from Figure 2C,D, respectively. The spectral peaks at the fundamental frequency (37 Hz) and the first harmonic frequency (74 Hz) are marked by dashed lines.

### 2.6. Full Informational Spectral Analysis of Source Activities Using Holo-Hilbert Spectral Analysis (HHSA)

In this study, we adopted the Holo-Hilbert spectral analysis (HHSA) [48] to analyze the spectral information of the ASSR source activities estimated by MNE. The HHSA utilizes a two-layer empirical mode decomposition (EMD) architecture to extract the frequency-modulated (FM) information and the amplitude-modulated (AM) information. In the HHSA, the first EMD was applied to decompose the oscillatory signal into a set of intra-wave frequency-modulated IMFs, denoted as IMF_FM_. The second EMD was applied to the upper envelope of each IMF_FM_ to obtain amplitude-modulated IMFs, denoted as IMF_AM_. The EMD was proposed by Huang et al. (1998) [49]. It attempts to sequentially decompose a signal into the sum of a finite number of intrinsic mode functions (IMFs) by iteratively conducting a sifting process, representing coarse-to-fine information of the recordings. Each IMF is a simple oscillatory signal whose amplitudes and frequencies are allowed to be varied with time so that it is beneficial to present local characteristics of nonstationary signals. The IMF is decomposed with the following definitions: (1) the number of local extrema (including local maxima and local minima) and the number of zero-crossings must either equal or differ at most by one, and (2) the mean value of the envelope defined by the local maxima and the envelope defined by the local minima are zeros.

After applying the first EMD, the ASSR source activities *x*(*t*) can be represented as the summation of IMF_FM_s, represented as:(6)$$x(t)=\sum_{j=1}^{K}F_{j}(t),$$
where $F_{j}(t)$ is the *j*th IMF_FM_.

Each IMF_FM_ has its characteristic carrier frequency with its own time-varying modulation function and can be expressed as follows:(7)Fj(t)=Re{aj(t)⋅eiφj(t)},
where Re{⋅} is the function for the real-valued signal, $a_{j}(t)$ is the real-valued modulation function for the *j*th IMF_FM_, $\varphi_{j}(t)=\int_{t}\omega_{j}(\tau )d\tau$ is the accumulated phase from the initial state to time *t*, and $\omega_{j}(\tau )$ is the instantaneous frequency at time *τ*.

By applying the second EMD to the envelope modulation function $a_{j}(t)$ of *j*th IMF_FM_, the $a_{j}(t)$ can be expressed as the summation of IMF_AM_s:(8)aj(t)=Re{∑m=1Maj,m(t)⋅eiψj,m(t)},
where $a_{j,m}(t)$ is the *m*th IMF_AM_ for *j*th IMF_FM_, $\psi_{j,m}(t)=\int_{t}\Omega_{j,m}(\tau )d\tau$ is the accumulated phase, and $\Omega_{j,m}(\tau )$ is the instantaneous frequency at time *τ* for the *m*th IMF_AM_.

The $\omega_{j}(t)$ and $\Omega_{j,m}(t)$ are the instantaneous frequencies of the *j*th IMF_FM_ in the first EMD and the *m*th IMF_AM_ in the second EMD, respectively. The instantaneous frequencies $\left(\omega_{j}(t),\Omega_{j,m}(t)\right)$ were incorporated with the magnitude of envelop function $\left|a_{j,m}(t)\right|$ across all time points into vectors and organized as a three-dimensional AM-FM matrix $\left({\vec{\omega }}_{j},{\vec{\Omega }}_{j,m},\left|{\vec{a}}_{j,m}\right|\right)$. The AM-FM matrices obtained from all IMFs were integrated together to achieve the Holo-Hilbert spectrum (HHS).

The signal processing steps for the HHSA processing are listed as follows:

Perform the first EMD to decompose the ASSR source activities into IMF_FM_s, and calculate instantaneous frequencies *ω*.Take the absolute values of the IMF_FM_s.Generate upper and lower envelopes for the absolute values of the IMF_FM_s by connecting the local extrema using spline interpolation.Perform the second EMD to the upper envelopes of the IMF_FM_s to obtain IMF_AM_s and calculate instantaneous frequencies Ω for IMF_AM_s.Arrange instantaneous amplitudes $\left|a_{j,m}(t)\right|$ and instantaneous frequencies *ω_j_*(*t*) and Ω*_j,m_*(*t*) into vectors to form a three-dimensional AM-FM matrix.Construct the Holo-Hilbert Spectrum (HHS) by integrating the three-dimensional AM-FM matrices from all IMF_FM_s and IMF_AM_s.

Figure 3A,B demonstrates the HHS plots of the LAC and RAC source activities shown in Figure 2C,D, respectively. The lower panels and the left panels in Figure 3A,B are the FM and AM spectra, which were calculated by accumulating the values on HHS across all AM and FM frequencies, respectively. It can be observed that the FM spectra (lower panels) have clear spectral peaks at the ASSR fundamental frequency and its first harmonic frequency (marked by blue and green dashed lines). For the AM spectra (left panels), since we are interested in the responses at stimulus-related frequencies, the AM spectra at 37 Hz (the fundamental frequency) and 74 Hz (the first harmonic frequency) are plotted and marked in blue and green colors, respectively. The AM spectra at 37 Hz have spectral peaks around 2, 4.6 and 9.2 Hz, in which the 9.2 Hz could be the harmonic frequency of 4.6 Hz. In contrast, the AM spectra at 74 Hz have spectral peaks around 5.6 and 13 Hz. Since the temporal changes in amplitudes can result in frequency splitting or spectral broadening on traditional Fourier spectra, the HHSA interprets the amplitude changes in AM spectrum on HHS and avoids the frequency splitting problem on its FM spectrum.

## 3. Results

In this study, we studied the FM and AM spectra in monaural stimulations using HHSA. The auditory-induced neural activities on the left and right auditory cortexes (LAC and RAC) were obtained by means of using the minimum norm estimation (MNE) technique, and then the HHSA was applied to analyze the FM and AM frequency responses. Figure 4 shows the cross-subject average of FM spectra over the 25 subjects during left and right ear stimulations. The 37 Hz amplitudes of left monaural stimulation in LAC and RAC were 2.19 ± 0.07 μv/Hz vs. 2.25 ± 0.1 μv/Hz, and the 37 Hz amplitudes of right monaural stimulation in LAC and RAC were 2.58 ± 0.06 μv/Hz vs. 2.49 ± 0.09 μv/Hz. The 37 Hz auditory responses in right-ear stimulation were all higher than the auditory responses in left-ear stimulation (*p* < 0.01; Wilcoxon signed rank test), and the auditory responses on the contralateral side were higher than the responses on the ipsilateral side (*p* < 0.05; Wilcoxon signed rank test). For the amplitudes at 74 Hz (first harmonic frequency), the amplitudes of left monaural stimulation in LAC and RAC were 1.02 ± 0.10 μv/Hz vs. 0.99 ± 0.13 μv/Hz, and the 74 Hz amplitudes of the right monaural stimulation in LAC and RAC were 1.19 ± 0.10 μv/Hz vs. 1.14 ± 0.09 μv/Hz. The amplitudes in right-ear stimulation were higher than the amplitudes in left-ear stimulation (*p* < 0.01; Wilcoxon signed rank test). The 74 Hz amplitude of RAC was higher than the amplitude of LAC in left ear stimulation (*p* < 0.05; Wilcoxon signed rank test).

The noteworthy benefit of HHSA is its capability to analyze AM spectrum for a measured signal. Figure 5 and Figure 6 demonstrate the AM spectra of the auditory responses at 37 and 74 Hz, respectively. The cross-subject averages of AM spectra at 37 Hz during left and right monaural stimulations are shown in Figure 5A,B, respectively. Both the left and right monaural stimulations presented three clear spectral peaks at 2.5, 5 and 9 Hz. The amplitudes of 2.5, 5 and 9 Hz of LAC in left-ear stimulations were 0.115, 0.118 and 0.083 μv/Hz, respectively, and the amplitudes of 2.5, 5 and 9 Hz of LAC in right-ear stimulations were 0.123, 0.124 and 0.084 μv/Hz, respectively. For the AM spectra of 37 Hz in right-ear stimulations, the amplitudes of 2.5, 5 and 9 Hz in LAC were 0.134, 0.143 and 0.100 μv/Hz, respectively, and the amplitudes of 2.5, 5 and 9 Hz in RAC were 0.132, 0.138 μv/Hz and 0.098 μv/Hz, respectively. The 2.5, 5 and 9 Hz were the three main peaks at the 37 Hz FM frequency. Since the amplitudes at 2.5 and 5 Hz had similar amplitude levels and the third peak (9 Hz) was not located at the second harmonic frequency of 2.5 Hz, the 5 and 9 Hz were not likely the harmonic frequency peaks of the 2.5 Hz component in each AM spectrum. The three modulation frequencies might be generated from distinct neural mechanisms. Further studies are needed to check the physiological meanings of these peaks on the steady-state auditory responses.

Examining the AM spectra of the 74 Hz auditory responses, three spectral peaks at 3 Hz, 6 Hz and one another spectral peak around 12 Hz were observed in the left-ear and right-ear stimulations. Similar to the three spectral peaks of AM spectra at 37 Hz (Figure 5), three spectral peaks at 3, 6 and 12 Hz were observed. The frequencies of the third spectral peaks in left-ear stimulations were 1 Hz higher than the third spectral peaks in right-ear stimulations. The amplitudes of the 3, 6 and 13 Hz in left-ear stimulations were 0.047, 0.037 and 0.026 μv/Hz in LAC, and the amplitudes were 0.050, 0.033 and 0.021 μv/Hz in RAC, respectively. For the AM spectra of 74 Hz in right-ear stimulations, the amplitudes of 3, 6 and 12 Hz were 0.055, 0.043 and 0.03 μv/Hz in LAC, and the amplitudes were 0.053, 0.039 and 0.027 μv/Hz in RAC, respectively. The 6 and 12 Hz could be the first and second harmonic frequencies of 3 Hz in the AM spectrum of 74 Hz auditory responses.

The HHSA enables scientists to observe the amplitude modulations in the measured signals. With the AM-FM representation of HHSA, the AM spectrum perfectly explains the effects of amplitude changes on each FM frequency, and the HHSA arranges the FM and AM components in a two-dimensional HHS representation (see Figure 3). In Figure 7, the Fourier spectra (Welch’s power spectrum [53]; signal window 1000 ms; overlapping 500 ms) (marked in blue) of source activities in LAC and RAC were overlapped with the FM spectra shown in Figure 4 (marked in red). For the 37 Hz fundamental frequency, it can be observed that the Fourier spectrum in LAC during the left-ear stimulation (the left panel in Figure 7A) had two additional spectral side peaks, located at 33.33 and 40.74 Hz, which were 3.67 and 3.74 Hz away from the 37 Hz stimulation frequency, respectively. The spectral side peaks were not seen in the FM spectrum obtained from HHSA. For the spectral analysis of source activities in RAC (the right panel in Figure 7A), the Fourier spectra had two spectral side peaks at 31.48 and 42.59 Hz in the left-ear stimulation, which were 5.52 and 5.59 Hz away from the 37 Hz stimulation frequency, respectively. Compared to the FM spectrum of left-ear stimulation in RAC, the spectral side peaks were also not seen. It validates the benefit of the AM-FM analysis using HHSA. The HHSA expands the traditional Fourier spectrum into a two-dimension HHS representation so that the effects of AM components are not confused with FM components. Accordingly, the frequency splitting (trigonometric product-to-sum formula) caused by amplitude modulation can be corrected. A similar observation was found in the right-ear stimulation (Figure 7B). The Fourier spectrum in LAC had a 40.74 Hz spectral side peak, and the Fourier spectrum in RAC had 34 and 40.74 Hz spectral side peaks. In contrast to the 74 Hz component on Fourier spectra, no 74 Hz spectral peak was found. The missing 74 Hz harmonic frequency could be blurred owing to the frequency splitting or spectral broadening caused by the effects of amplitude modulation. In addition, an 80.5 spectral side peak was observed in all FM spectra, which might require further studies to understand its physiological meaning.

## 4. Discussion

In the present study, the ASSR signals in human brain were analyzed using the combination of CCA and HHSA. The CCA was adopted to extract ASSR-related signal features, and the selected canonical vectors were used to reconstruct noise-suppressed auditory responses. The HHSA was applied to analyze the AM-FM spectral information by calculating the instantaneous frequencies (IF) in local time slots. However, the IF requires the calculation of the first derivative in signal phases, which is sensitive to noise when the phase difference between samples is too large. Accordingly, the adoption of CCA in the preprocessing step helps the signal frequencies to be confined around our interested ASSR frequencies so that the phase fluctuations can be suppressed. Since ASSR is a sinusoidal-like wave with frequencies located at the stimulation frequency and its harmonics, the adoption of CCA can effectively enhance the ASSR-related features by setting the sinusoidal waves at fundamental and harmonic frequencies. The CCA aims to find a linear transformation to maximally correlate the extracted EEG features with the preset sinusoidal vectors. The canonical vectors in *U* (see Equation (3)) constructed a feature space. The multichannel EEG samples were projected to these canonical vectors for the extraction of ASSR-related signals. After calculating the weights of these canonical vectors on each EEG channel, the noise-suppressed EEG data can then be reconstructed by the summation of multiplying the weights of ASSR-related canonical vectors with their time series (see Equation (5)).

In our study, the canonical vectors with a statistical significance level of *p* < 0.01 were identified as ASSR-related canonical vectors for reconstructing ASSR-related signals. Compared to other CCA papers [54], the task-related canonical vectors were determined by defining a fixed threshold level on the correlation values. Since EEG signals are weak and susceptible to environmental noises, e.g., electromagnetic interference and motion artifacts, the use of a fixed correlation threshold for component selection might be risky owing to the contamination of nonstationary noise in varied measurement environments. For example, the correlation values of the canonical vectors extracted from right auditory ROI were 0.032, 0.031, 0.004, and 0.003 (see Figure 1B), respectively, and the correlation values of the canonical vectors extracted from left auditory ROI were 0.033, 0.017, 0.010, and 0.003 (see Figure 1C), respectively. For the canonical vectors in Figure 1C, the correlation values of the second and third canonical vectors (0.017 and 0.010) had a gap lower than the correlation value of the first canonical vector (0.033) and a gap higher than the correlation value of the fourth canonical vector (0.003). This resulted in the difficulty of finding a clear threshold for selecting the ASSR-related canonical vectors since the correlation values are changed depending on the noise patterns and noise magnitudes exerted on the ASSR responses. Determining the correlation threshold based on unsupervised clustering techniques, such as k-means, fuzzy C-means, Gaussian mixture model, etc. [55], could still be heuristic because the cluster number and grouping strategy should be defined by users in those unsupervised cluster classifiers. Setting wrong parameters in the use of clustering methods could include task-unrelated components and result in feature loss or noise engagement. Therefore, in order to find pertinent canonical vectors for reconstructing ASSR signals, we created a spatial map for each canonical vector by projecting the canonical vector on the measured signal of each EEG channel. We found the canonical vectors with statistical significance levels had clear spatial distribution concentrated around the left or right auditory ROI, which demonstrated the source activities of these canonical vectors originated from the left or right auditory cortexes. The canonical vectors with significant *p* values were thence chosen for reconstructing ASSR signals.

Because EEG is more sensitive to the radial component of neuroelectric activity, the measured EEG amplitude is decreased owing to the projection angle between the dipole orientation and the radial direction of the electrode surface [56]. Therefore, current sources with rotating orientations can result in amplitude changes in the electric fields picked up by the EEG sensor [56]. In this study, the HHSA was applied to the neural activities in the auditory cortex to avoid the influence of source orientations. The source activities were first estimated using minimum norm estimation (MNE) [51], and the neural activities at LAC and RAC were used as input signals for HHSA analysis in order to avoid the influence of rotating source caused amplitude changes on the sensor level (EEG sensors).

One noteworthy advantage of HHSA is the extending of the traditional spectral analysis from FM to AM-FM representation [57,58,59]. By unfolding the amplitude modulation and carrier frequency into AM and FM axes, the two-dimension frequency expression resolves the frequency splitting problem caused by the trigonometric product-to-sum formula. It has been reported that different physiological states can affect the modulation of rhythmic source activities. Because the human brain is a dynamic system, neurophysiological activities can be modulated by attention [6,7,10,60], emotional state [61,62,63], autonomic regulation [64,65,66], brain resting-state network [67,68,69], etc. In our study, we found the ASSRs at 37 and 74 Hz were modulated by AM waves with composite frequency components. For the 37 Hz auditory responses, the AM spectra had three spectral peaks (see Figure 5), located around 2.5, 5 and 9 Hz, and a broad frequency modulation range with −3 dB bandwidth about 20 Hz (±10 Hz around 37 Hz). Similarly, the AM spectrum of the 74 Hz auditory responses presented three peaks (see Figure 6), located around 3, 6 and 12 Hz, and a broad modulation frequency range with −3 dB bandwidth of about 14 Hz (±7 Hz around 74 Hz). As shown in Figure 7, the spectral side peaks were observed around 37 Hz in Fourier spectra, which could be the splitting frequencies caused by the product-to-sum formula. The amplitude modulation results in spurious spectral peaks or broadened spectrum, which could result in misleading or wrong interpretation of the measured signal. In Figure 7, the first harmonic frequency (74 Hz) was missing in all the Fourier spectra and restored in the FM spectra obtained from HHSA, which demonstrated the effectiveness of HHSA in avoiding modulation-caused spectral broadening.

## 5. Conclusions

The present study proposed a CCA and HHSA combination method for ASSR analyses. With the benefit of HHSA in AM and FM frequency analysis, the information on AM and FM spectra can be obtained. Because the frequency analysis in HHSA is calculated based on the IF technique, which is sensitive to noise, we adopted CCA as a preprocessing tool to avoid fluctuations in the estimated instantaneous frequencies. With the HHSA, we have demonstrated the existence of modulation frequencies in the measured ASSR responses. Both the 37 and 74 Hz auditory responses were modulated by AM spectra with at least three composite frequencies. In contrast to the frequency splitting in traditional Fourier spectra, the HHSA interprets the time-varying amplitude changes by AM frequencies on HHS. The proposed method effectively corrects the frequency splitting caused by modulation effects (product-to-sum formula), which avoids a misleading or wrong interpretation of the spectral analysis in the measured steady-state responses.

## Figures and Tables

**Figure 1 jcm-11-03868-f001:**
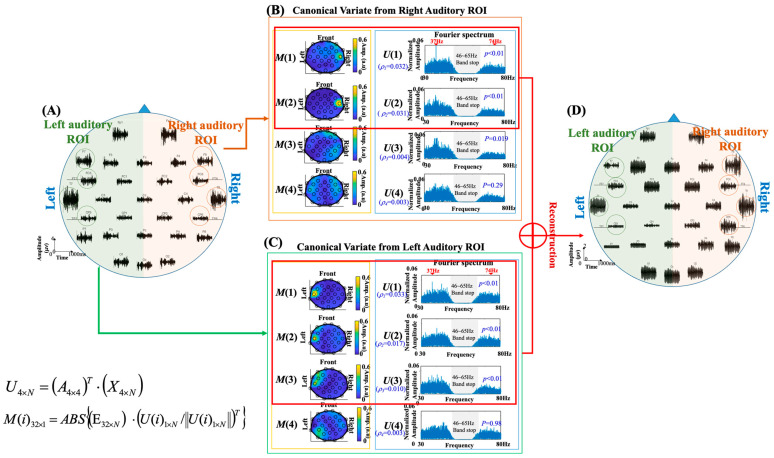
The demonstration of CCA for ASSR extraction in subject5: (**A**) The left panel shows the channel plot of the recorded EEG signals. (**B**) The spatial maps and Fourier spectra of canonical vectors by setting the COI located in the vicinity of the right auditory cortex. (**C**) The spatial maps and Fourier spectra of canonical vectors by setting the COI located in the vicinity of the left auditory cortex. (**D**) The channel plot of the ASSR signals were reconstructed from ASSR-related canonical vectors.

**Figure 2 jcm-11-03868-f002:**
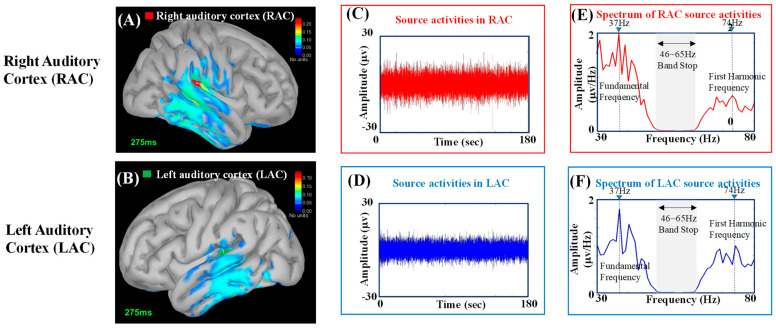
The source activities of ASSRs estimated by MNE in RAC and LAC: (**A**) The neural sources of ASSR in RAC; (**B**) the neural sources of ASSR in LAC; (**C**) the source activities in RAC; (**D**) the source activities in LAC; (**E**) the Fourier spectrum of the source activities in RAC; (**F**) the Fourier spectrum of the source activities in LAC.

**Figure 3 jcm-11-03868-f003:**
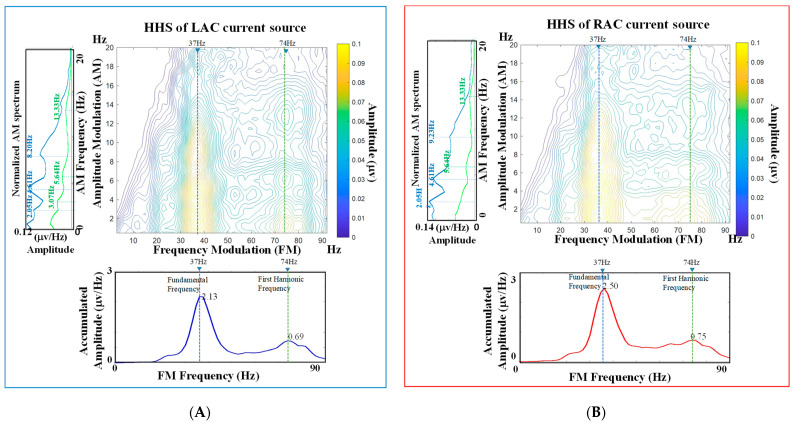
The HHS plots of the source activities in auditory cortexes: (**A**) The HHS plot of the source activities in LAC. The lower panel shows the FM spectrum, and the left panel shows the AM spectrum. (**B**) The HHS plot of the source activities in RAC. The lower panel shows the FM spectrum, and the left panel shows the AM spectrum.

**Figure 4 jcm-11-03868-f004:**
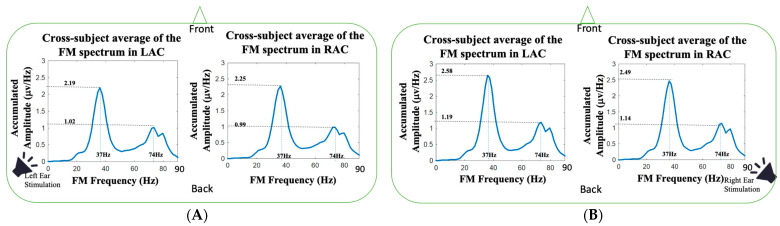
The FM spectra averaged across the twenty-five subjects in our study: (**A**) The cross-subject average of FM spectrum in LAC and RAC during left-ear auditory stimulations; (**B**) the cross-subject average of FM spectrum in LAC and RAC during right-ear auditory stimulations.

**Figure 5 jcm-11-03868-f005:**
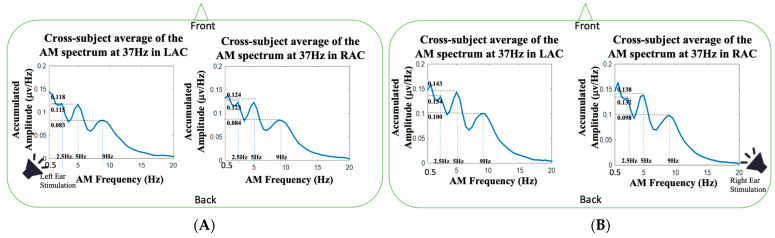
The AM spectra of the 37 Hz FM frequency averaged across the twenty-five subjects in our study: (**A**) the cross-subject average of the AM spectrum at 37 Hz in LAC and RAC during left-ear auditory stimulations; (**B**) the cross-subject average of the AM spectrum at 37 Hz in LAC and RAC during right-ear auditory stimulations.

**Figure 6 jcm-11-03868-f006:**
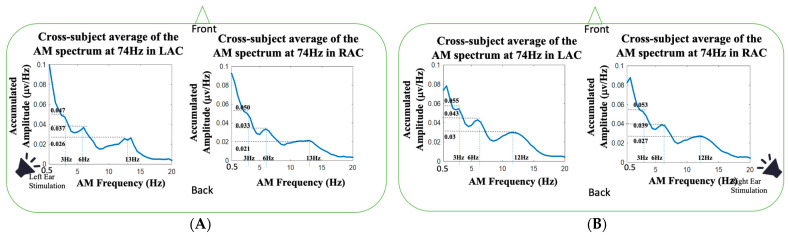
The AM spectra of the 74 Hz FM frequency averaged across the twenty-five subjects in our study: (**A**) the cross-subject average of the AM spectrum at 74 Hz in LAC and RAC during left-ear auditory stimulations; (**B**) the cross-subject average of the AM spectrum at 74 Hz in LAC and RAC during right-ear auditory stimulations.

**Figure 7 jcm-11-03868-f007:**
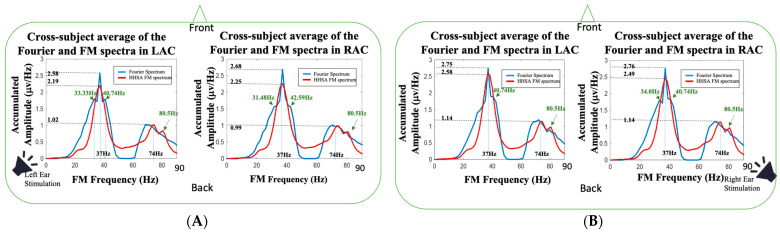
The comparison of the Fourier spectra (marked in blue) and the FM spectra (marked in red) obtained from left-ear and right-ear auditory stimulations: (**A**) the cross-subject averaged spectra obtained from Fourier analysis and the HHSA during left-ear stimulations; (**B**) the cross-subject averaged spectra obtained from Fourier analysis and the HHSA during right-ear stimulations.

## Data Availability

Not applicable.

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
