# Peer review of "The Full Informational Spectral Analysis for Auditory Steady-State Responses in Human Brain Using the Combination of Canonical Correlation Analysis and Holo-Hilbert Spectral Analysis"

_jcm, 2022, doi:10.3390/jcm11133868_

Round 1

Reviewer 1 Report

Review different coding algorithm for data transmission in neural network, i.e., rate coding, temporal coding. The authors can use the following Paper:Capacity and error probability analysis of neuro-spike communication exploiting temporal modulation

Author Response

Authors appreciate reviewer’s comment. The review of different neural coding hypotheses has been added in the revised manuscript. One paragraph has been modified and the reference “Capacity and error probability analysis of neuro-spike communication exploiting temporal modulation” has been added. 

Reviewer 2 Report

This paper proposes a new spectral analysis for auditory steady-state responses. The proposed method using the combination of canonical correlation analysis and holo-Hilbert spectral analysis is better than Fourie spectral analysis.  Unfortunately, there are some unclear points as follows: Please revise them. 

  1. Please describe the correlation value ρi (i=1,2,3,4) for Figure 1. Which canonical vectors were eliminated for reconstructed responses ?
  2. Please add (a), (b) and (c) in Figure 1. (see line 268).
  3. Please explain how to get source activities in Figure 2 (c) and (d).
  4. Please add the reference to Section 2.6.
  5. Please explain how to decompose the oscillatory signal into a set of intra-wave frequency-modulated IMFs.
  6. Please explain the reason for the following sentences: the line 350-352 and line 355-356.  Are these differences statistically significant ?
  7. In Figures 5 and 6,  are three modulation frequencies meaningful/valid ?  Please explain the reason.

Author Response

Authors appreciate reviewer's comments. We have done a point-to-point revision in our revised manuscript. 

Reviewer 3 Report

The authors studied human ASSR using the combination of canonical correlation analysis and holo-hilbert spectral analysis. It was an interesting and important study as it may help improve the accuracy of the test. The gap of knowledge and objectives has been well described in the introduction. The methodology was written in detail. The conclusion was well supported by the results.

Author Response

Authors appreciate reviewer's comment. Thank you. 

Round 2

Reviewer 2 Report

Nothing

This manuscript is a resubmission of an earlier submission. The following is a list of the peer review reports and author responses from that submission.